# Interactions among temperature, moisture, and oxygen concentrations in controlling decomposition rates in a boreal forest soil

Carlos A. Sierra[1], Saadatullah Malghani[1], and Henry W. Loescher[2,3]

[1]Max Planck Institute for Biogeochemistry, Hans-Knöll Str. 10, 07745 Jena, Germany
[2]Battelle-National Ecological Observatory Network, 1685 38th St., Boulder, CO 80301, USA
[3]Institute of Alpine and Arctic Research (INSTAAR), University of Colorado, Boulder, CO 80309, USA

*Correspondence to:* Carlos A. Sierra (csierra@bgc-jena.mpg.de)

**Abstract.** Determining environmental controls on soil organic matter decomposition is of importance for developing models that predict the effects of environmental change on global soil carbon stocks. There is uncertainty about the environmental controls on decomposition rates at temperature and moisture extremes, particularly at high water content levels and high temperatures. It is uncertain whether observed declines of decomposition rates at high temperatures are due to declines in the heat capacity of extracellular enzymes as predicted by thermodynamic theory, or due to simultaneous declines in soil moisture. It is also uncertain whether oxygen limits decomposition rates at high water contents. Here we present results from a full factorial experiment using organic soils from a boreal forest incubated at high temperatures (25 and 35 degrees C), a wide range of water-filled pore space WFPS (15, 30, 60, 90%), and contrasting oxygen concentrations (1 and 20%). We found support for the hypothesis that decomposition rates are high at high temperatures provided enough moisture and oxygen is available for decomposition. Furthermore, we found that decomposition rate is mostly limited by oxygen concentrations at high moisture levels; even at 90% WFPS, decomposition proceeded at high rates in the presence of oxygen. Our results suggest an important degree of interactions among temperature, moisture, and oxygen in determining decomposition rates at the soil-core scale.

## 1 Introduction

The physical environment has a strong control on soil organic matter dynamics by modulating the rates of biological activity and therefore the rates at which organic matter decomposes. Hence, environmental change produced by global warming or changes in land use, can significantly affect organic matter decomposition rates and the capacity of soils to store carbon (Trumbore, 1997; Schlesinger and Andrews, 2000; Davidson and Janssens, 2006; Luo et al., 2016).

Among different environmental factors, temperature, moisture, and oxygen levels in soils have a strong control on the rate of soil organic matter (SOM) decomposition (Greenwood, 1961; Bunnell et al., 1977; Swift et al., 1979; Skopp et al., 1990; Davidson et al., 2012; Moyano et al., 2013). Yet, there are still large uncertainties in our understanding on how to model environmental controls on decomposition rates. For instance, many different functions have been previously proposed

to represent environmental controls on decomposition rates, most functions disagree at the extremes of the temperature and moisture ranges, and it is difficult to select appropriate functions due to large uncertainties in available data (Sierra et al., 2015b).

In particular, there is uncertainty about the shape of decomposition functions at high temperature levels. Traditional Arrhenius kinetics predict that rates of decomposition increase monotonically as temperature increases (Sierra, 2012), a behavior
well supported by classical thermodynamic theory, and in particular by its second law (Reif, 2009). However, an important number of biochemical studies shows that at a certain temperature threshold, generally above 45°C, enzymes denature and lose their capacity to catalyze reactions, slowing down rates of substrate consumption (Fields, 2001). Hobbs et al. (2013) and Schipper et al. (2014) suggest that this temperature limit for enzyme denaturation may be too high to be relevant in soils, and propose an alternative thermodynamic theory that predicts a lower temperature threshold when decomposition rates reach a
maximum. Their macromolecular rate theory (MMRT) is based on the idea that the activation energy in the Arrhenius equation is temperature dependent and related to negative changes in the heat capacity of enzyme-catalyzed reactions.

Both enzyme denaturalization and MMRT operate at the macromolecular enzyme-substrate level, and assume that other environmental factors such as moisture remain constant as temperature increases. At larger spatial and temporal scales though, multiple reactions occur simultaneously at different rates, and different environmental factors interact with temperature. For
instance, there is large empirical and theoretical evidence showing that interactions with soil moisture lead to strong changes in decomposition rates not predicted by changes in temperature alone (Bunnell et al., 1977; Davidson and Janssens, 2006; Sierra et al., 2015b; Tucker and Reed, 2016; Zhou et al., 2016).

Soil moisture plays two contrasting roles as a modulator of decomposition rates. On the one hand, soil water solubilizes substrates and increase their availability in active microbial sites through diffusion. As soil water increases, it also reduces
physiological stress on microbes by reducing soil matric potential (Moyano et al., 2013; Manzoni et al., 2014). On the other hand, as moisture increases it fills up available pore spaces and reduces oxygen levels necessary for aerobic microbial activity (Skopp et al., 1990; Moyano et al., 2013). Oxygen exerts an important control on the speed of aerobic decomposition for its role as an electron acceptor in the mineralization of SOM (Greenwood, 1961; Keiluweit et al., 2016). As moisture increases in soils, aeration and oxygen levels inevitably decrease.

Progress in understanding multiple-factor effects on SOM decomposition has been hindered by a paucity of experimental research (Dieleman et al., 2012; Leuzinger et al., 2011; Zhou et al., 2016). Full factorial experiments with multiple factors and levels are rare, even though they provide basic understanding on the independent and combined effects of environmental factors on decomposition. Furthermore, there has been little work studying how multiple environmental factors affect multiple decomposition rates that account for the heterogeneity of substrates and processes in soils.

Here, we use a full-factorial incubation experiment in combination with model-data integration to address the questions: i) do decomposition rates remain high at high temperatures provided moisture and oxygen are not limiting?, ii) do decomposition rates remain high at high moisture levels provided oxygen and temperature are not limiting? These questions are important because they provide insights about the best possible model structures required to represent SOM decomposition at extreme environmental conditions, and in light of global environmental change.

## 2 Methods

### 2.1 Soils and incubation experiment

We developed a full factorial incubation experiment with the manipulated treatments being temperature (25, 35 °C), soil water content (15, 30, 60 90% water-filled pore space), and oxygen concentration in the pore space (1 and 20%), with soils enclosed in PVC cylinders (10 cm diameter and 20 cm height) containing in about half of their volume 450 g of homogenized soil. The approximate bulk density within each cylinder was $0.6$ g cm$^{-3}$. Organic soil was collected from the A horizon of a boreal forest dominated by black spruce at the Caribou Poker watershed in central Alaska, USA (65° 9' 21.365" N, 147°, 29', 28.74" W). The soil is classified as a *Histic Pergelic Cryaquept* in a Gilmore silt loam series from the United States Department of Agriculture Natural Resource Conservation Service system. It has a depth of 1 m followed by permafrost, and a water table depth of 20 cm. Soil temperatures in the active layer fluctuate annually from a minimum of -19 to a maximum of 18°C. The carbon (C) content of a subsample of the soil used for incubations was $46.9 \pm 0.1$ mg C g$^{-1}$ soil. We chose a boreal soil for this experiment because its high organic matter content may avoid potential substrate limitations during incubations, and the low temperatures at which its microbial community is constantly exposed facilitates the possibility of observing strong responses at the extreme of the temperature range. In addition, soils in the boreal region are experiencing fast changes in environmental variables, so it is of high relevance to study this type of systems.

This soil is identical as the one used in a companion paper (Sierra et al., 2015a), with the exception that in that publication we used only one single treatment to illustrate results from an identifiability analysis, while here we report data from the complete full factorial experiment.

Prior to the incubations, the soil was homogenized, passed through a 2 mm sieve, and large roots (> 2 mm diameter) were removed. Four replicates per treatment were placed in two climate chambers at a constant temperature each. The bottom of each column was connected to an air inlet system that continuously flushed soil columns from the bottom at a rate of $30 \pm 3$ ml min$^{-1}$ with air of known oxygen (1 or 20 %) and $CO_2$ (350 ppm) concentration. The headspace exiting each column (after passing through the soil) was connected to an automated multiport stream selection valve, and then analyzed for $CO_2$ using an infrared gas analyzer (LI-6262 LI-COR Inc., Lincoln, USA). Respiration rates, measured as $CO_2$ production fluxes, were calculated as the difference in $CO_2$ concentration between the fluxes in the outlet and inlet streams, multiplied by the air mass flow rate. Moisture loss ($\sim$1 g per day per cylinder) due to continuous flushing of dry air was compensated by adding water to replace lost of mass once every week. Additional details about the system can be found in Malghani et al. (2013).

### 2.2 Statistics and model optimization

Treatment means of total respired $CO_2$ from the 35-day incubation period (total sum for each cylinder) were compared using analysis of variance $F$-statistic. We used a linear fixed-effects model using as independent variables the three independent treatments as well as their combination.

To evaluate the effect of the different treatments on decomposition rates, we used a simple two-pool model. In a previous analysis, we found that for incubation data no more than 3 or 4 parameters can be optimized simultaneously without encoun-

tering identifiability (equifinality) problems (Sierra et al., 2015a). When the number of parameters to identify is large and the number of observations low, the identifiability problem results in collinearity of the parameters. This means that changes in the value of one parameter can be compensated by changes in the value of another parameter without any effect in predicting the observed data. In these cases, multiple parameter sets predict equally well the data, and it is not possible to uniquely identify the best underlying mechanisms that explain the observations (Soetaert and Petzoldt, 2010; Sierra et al., 2015a). For this reason, we chose a simple model that has three main parameters and is expressed as

$$\frac{d\boldsymbol{C}}{dt} = \xi \cdot \begin{pmatrix} -k_1 & 0 \\ 0 & -k_2 \end{pmatrix} \cdot \begin{pmatrix} C_1 \\ C_2 \end{pmatrix}; \quad \boldsymbol{C_0} = C_0 \cdot \begin{pmatrix} \gamma \\ 1-\gamma \end{pmatrix}, \tag{1}$$

where the amount of C in the system (in grams) is stored in a fast and a slow pool $C_1$ and $C_2$, with corresponding decomposition rates $k_1$ and $k_2$ (in days$^{-1}$). The initial amount of carbon in the system $C_0$ is partitioned according to a proportion $\gamma$, and the environmental term $\xi$ is a product of three functions that depend on the environment $f(T)$, $f(M)$, and $f(O)$ such that

$$\xi = f(T) \cdot f(M) \cdot f(O) = Q_{10}^{\frac{T-10}{10}} \cdot \frac{M}{K_M + M} \cdot \frac{O}{K_O + O}, \tag{2}$$

where $K_M$ and $K_O$ are half-saturation constants for the soil moisture $M$ and soil oxygen concentration $O$ terms. Notice that equation (2) is a simplified version of the DAMM model of Davidson et al. (2012), and it is incorporated into a model (equation 1) that tracks the temporal dynamics of a fast and a slow pool simultaneously.

The model was solved numerically using the SoilR package (Sierra et al., 2012), and the total cumulative respiration flux was calculated from the numerical output as an integral (area under the curve) for the 35-day incubation period. We optimized two versions of the model of equation (1) to the observed data from the experiment using a Bayesian approach (Soetaert and Petzoldt, 2010). First, we optimized parameters of each treatment independently and setting $\xi = 1$. In this way we can observe possible trends in the parameters as a function of the environmental variables. Second, we pooled data from all treatments together and fitted the full model with $\xi$ expressed as in equation (2).

All analyses were performed in R (The R Foundation for Statistical Computing, Vienna), and all code and data to reproduce our results are available as supplementary material.

## 3  Results

Total respired $CO_2$ after 35 days of incubation showed a strong treatment effect for the three main variables ($F$-statistic $p$-value $< 0.001$ for the main treatment effects). Interactions among all treatment levels showed statistically significant effects ($F$-statistic $p$-value $= 0.0505$) suggesting that $CO_2$ efflux for this soil responded to different combinations of the treatment levels (Figure 1). A statistically significant interaction ($F$-statistic $p$-value $< 0.001$) was also found between the soil moisture and oxygen treatments. The largest amount of respired $CO_2$ was observed at the treatment with the highest temperature, moisture and oxygen levels (35°C, 90%, 20%), while the lowest amount was observed at the treatment with the lowest values for these

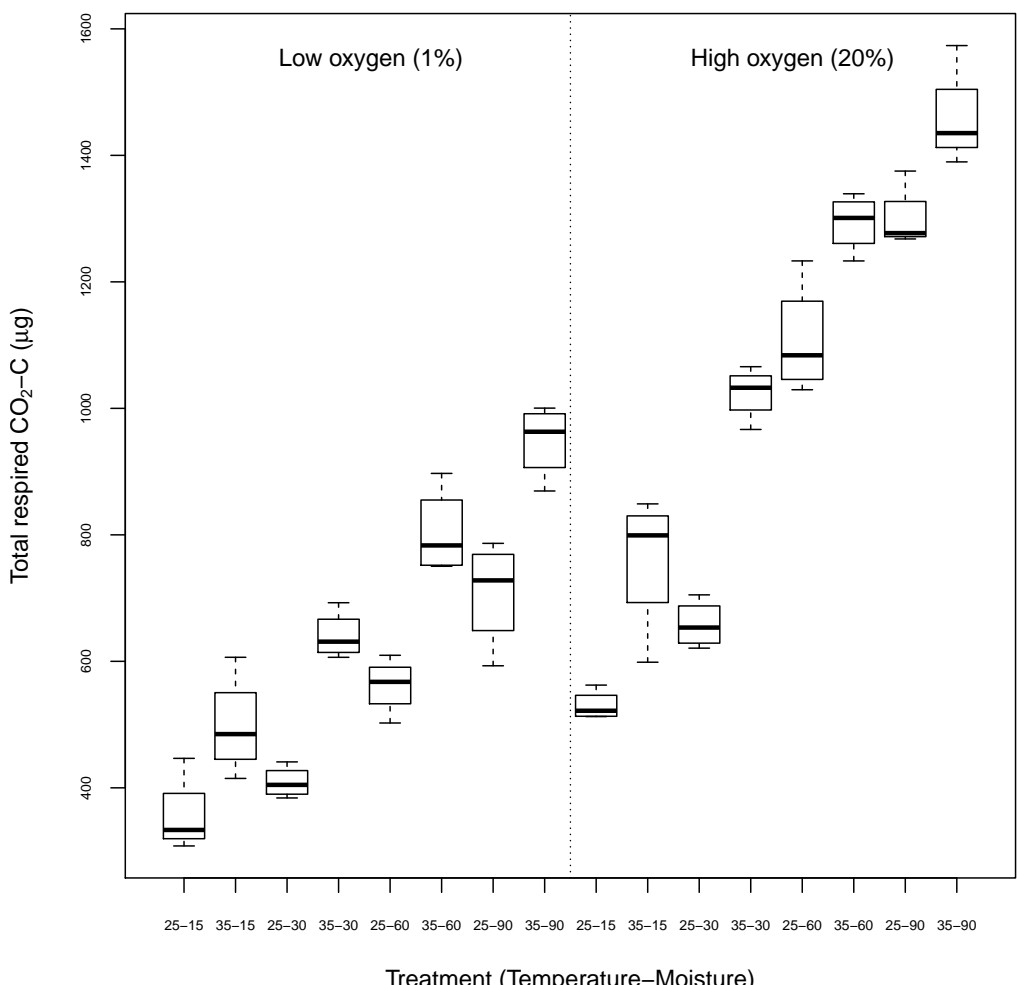

**Figure 1.** Total respired $CO_2$ integrated over the length of the experiment by treatment. Numbers in the treatment level represent the level of temperature (degrees Celsius) and soil water content (%).

variables (25°C, 15%, 1%), confirming that these three environmental variables collectively exert a strong and significant control on $CO_2$ production. Total respired $CO_2$ during the experiment did not decrease at the higher temperature level.

The results of the first model optimization showed that temperature generally increased decomposition rates of both fast and slow pools at similar moisture and oxygen levels, but uncertainties were generally large (Figure 2). Under higher temper-
5  atures, we also observed a larger proportion of carbon being mineralized faster and contributing to the initial respiration pulse (parameter $\gamma$). At lower oxygen levels, decomposition rates were slower than in similar treatments with higher oxygen levels.

Although we estimated only three parameters, there were already problems with convergence and identifiability in this optimization (cf. Sierra et al., 2015a), which means that the obtained values of some parameters can be compensated by

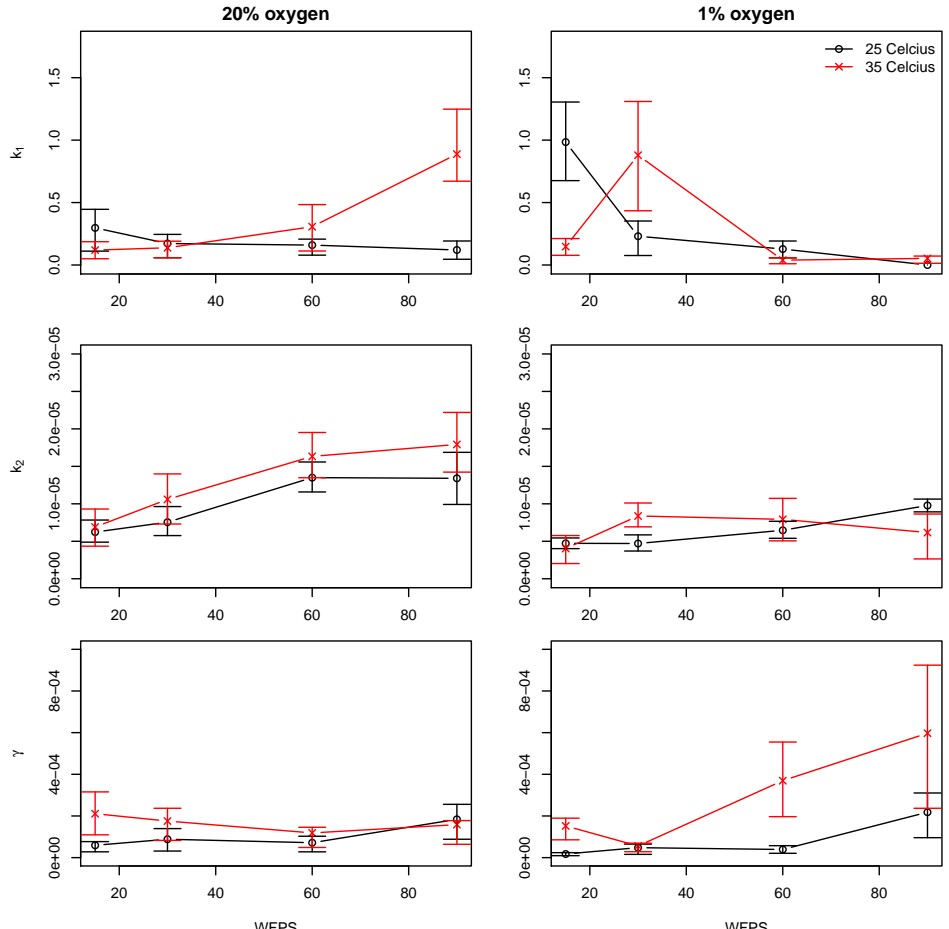

**Figure 2.** Results from the first model optimization procedure for the two pool model applied to each experimental treatment independently. Parameters optimized were $k_1$: decomposition rate of fast pool, $k_2$: decomposition rate of slow pool, $\gamma$: fraction of the total initial carbon in the fast pool. The experimental treatments were water-filled pore space WFPS (15, 30, 60, 90 %), oxygen concentration (1, 20%), and temperature (25, 35° Celcius). Arrows represent 25 and 75% quantiles of the distribution of the parameters obtained through Bayesian optimization.

proportional changes in the values of other parameters. In some cases, the optimization method also failed to converge to stable posterior distributions for some specific treatment combinations. The second optimization with the full dataset reduced these collinearity and convergence problems.

    The optimization of the full dataset did not provide evidence of strong collinearity as indicated by the low correlations
5   among posterior values (Figure 3). These obtained posterior values can then be summarized by simple statistics such as their mean, standard deviation (SD) and interquartile ranges (Table 1). In particular, the obtained values for $\gamma$, $k_1$, and $k_2$ indicate reference values for the partitioning coefficient and the decomposition rates under no treatment effects, i.e. $\xi = 1$.

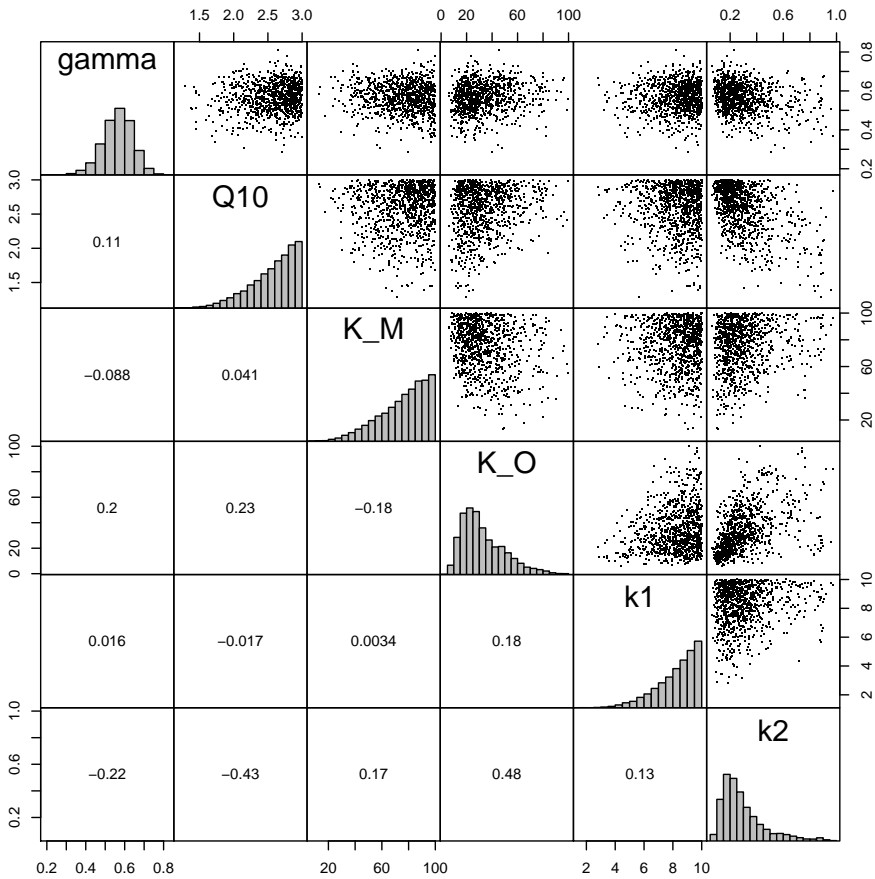

**Figure 3.** Posterior parameter values from the Bayesian optimization using the full model of equation (2). To avoid cluttering of the figure, only 1000 randomly samples values per posterior parameter set are plotted.

Using the obtained mean values of the posteriors with their respective 5-95% uncertainty ranges, we calculated and plotted the response functions $f(X)$ with their intrinsic sensitivities $\partial f(X)/\partial X$, particularly for the treatment levels used in our experiment (Figure 4). The optimized functions predict that decomposition rates increases as temperature, moisture and oxygen increases. For temperature, both the dependence and the sensitivity functions increase with temperature as well as the

5  uncertainty in the predictions. For moisture, as the water filled pore space increases, decomposition rates also increase, but their intrinsic sensitive declines. Similarly, as oxygen levels increase, decomposition rates are predicted to increase but their sensitivity is expected to decline (Figure 4). The obtained functions also suggest larger sensitivities of decomposition rates with respect to temperature than with respect to moisture or oxygen.

Combining together the optimized functions into the term $\xi = f(T) \cdot f(M) \cdot f(O)$, we obtained the interacting effect of treat-

10  ment level combinations on decomposition rates (Figure 5). Temperature, moisture and oxygen acted synergistically increasing

**Table 1.** Summary statistics of obtained posterior parameter values for the full model of equations (1) and (2) using a Bayesian optimization procedure. Quantiles at 5 and 95% level are reported as $q_5$ and $q_{95}$, respectively.

| Parameter | Units | Mean | SD | $q_5$ | $q_{95}$ |
|---|---|---|---|---|---|
| $\gamma$ | Proportion, unitless | 0.565 | 0.077 | 0.434 | 0.686 |
| $Q_{10}$ | Unitless | 2.574 | 0.33 | 1.928 | 2.971 |
| $K_M$ | % | 76.026 | 17.887 | 41.298 | 98.209 |
| $K_O$ | % | 33.157 | 17.11 | 12.340 | 66.926 |
| $k_1$ | day$^{-1}$ | 8.25 | 1.411 | 5.455 | 9.891 |
| $k_2$ | day$^{-1}$ | 0.293 | 0.163 | 0.117 | 0.642 |

decomposition rates in our experiment. Decomposition rates were twice as large at the highest treatment levels of temperature, moisture, and oxygen (35°C, 90%, 20%) than at the reference level ($\xi = 1$, $k_1$ and $k_2$ as in Table 1). For the lowest treatment levels (25°C, 15%, 1%), decomposition rates were reduced by a factor of 0.02 (two percent) from the reference level. Interactions between temperature and moisture in increasing decomposition rates were stronger when oxygen levels were high, but even at low oxygen levels increases in temperature and moisture resulted in small increases in decomposition rates.

## 4 Discussion

The statistical comparison of the obtained respiration data as well as the results from these two modeling exercises demonstrated strong interactions among three main environmental factors that control decomposition. The factorial nature of our experiment allowed us to calculate intrinsic sensitivities for these three environmental factors. Moreover, without controlled conditions, the effects of one variable would have been confounded by others. For example, increases in temperature are often accompanied by decreases in soil moisture, and increases in moisture are generally accompanied by decreases in soil oxygen concentrations. Our experimental design, with a continuous flow of oxygen through the soil column, helped us to control oxygen concentrations independent on moisture, which avoided possible confounding effects.

Our results support previous work on the control of these three environmental variables on decomposition (Bunnell et al., 1977; Davidson et al., 2012, 2014). In particular, our results show that decomposition rates at the soil-core scale are strongly controlled by an interaction among three main environmental variables that generally change in concert with one another in the natural soil environment.

Tucker and Reed (2016) showed that the interaction between temperature and moisture play an important role for predicting soil respiration rates in dry soils. Similarly to our study, these authors did not find a decline in soil respiration rates at high temperatures. But rather, they found a strong interaction between an exponential function for temperature effects and a moisture function that reached a maximum at high moisture levels. This lack of decline of the moisture function is expected for dry soils that do not reach water saturation levels. The higher moisture range covered in our study shows more clearly that there is a

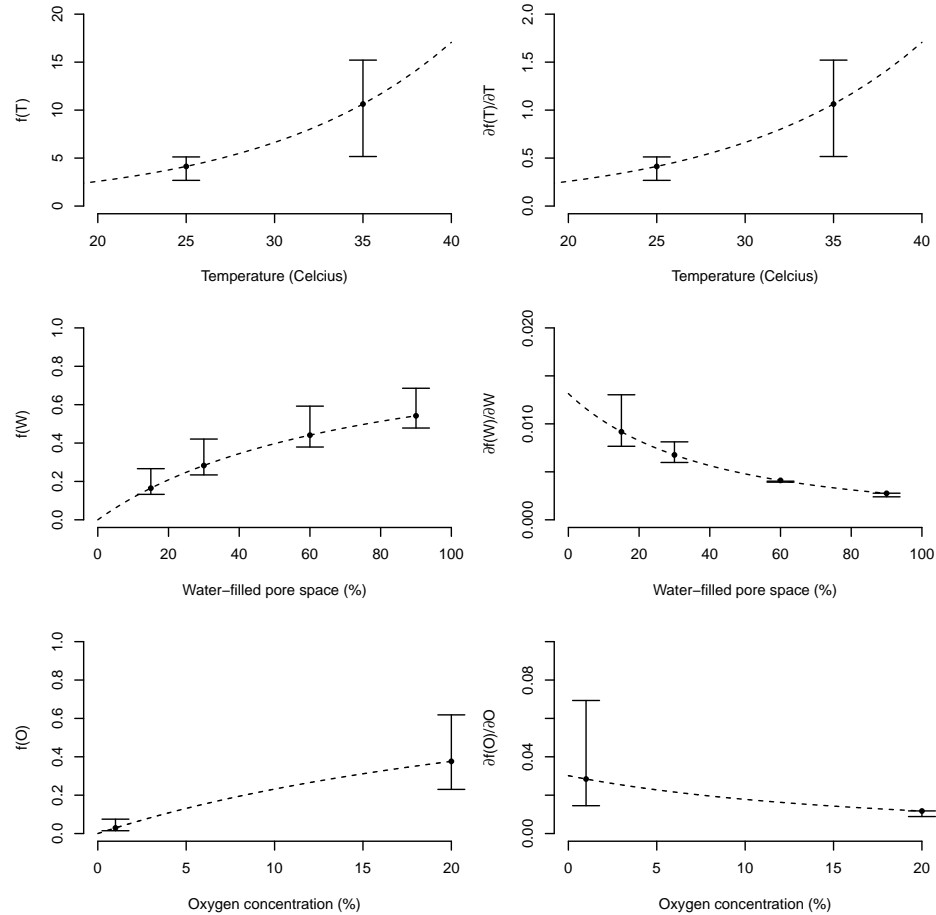

**Figure 4.** Shape of the response functions $f(T)$, $f(M)$, and $f(O)$ (dashed lines) calculated with the mean values of the posterior parameters, and their respective sensitivities $\partial f(T)/\partial T$, $\partial f(M)/\partial M$, and $\partial f(O)/\partial O$. Predictions for the treatment levels used in the experiment are presented as points with their respective 5-95% uncertainty.

decline at high moisture levels and it is mostly driven by oxygen availability. One limitation of our study however, was the use of only two levels for the temperature and oxygen treatments, from which it is difficult to derive specific trends. This is a natural limitation of full factorial experiments, where the addition of extra treatment levels considerably increases logistical challenges.

5    Our modeling exercise, particularly the optimizations with the full dataset, helped us to better understand the combined effect of temperature, moisture, and oxygen in modifying decomposition rates for the soil we studied. The intrinsic sensitivity of decomposition with respect to temperature increased with temperature, but the intrinsic sensitivities with respect to moisture and oxygen decreased with increasing levels of these variables (Figure 4). Since there are complex interactions among these three variables (Figure 5), specific responses due to their combined changes can only be predicted with the help of models. Al-

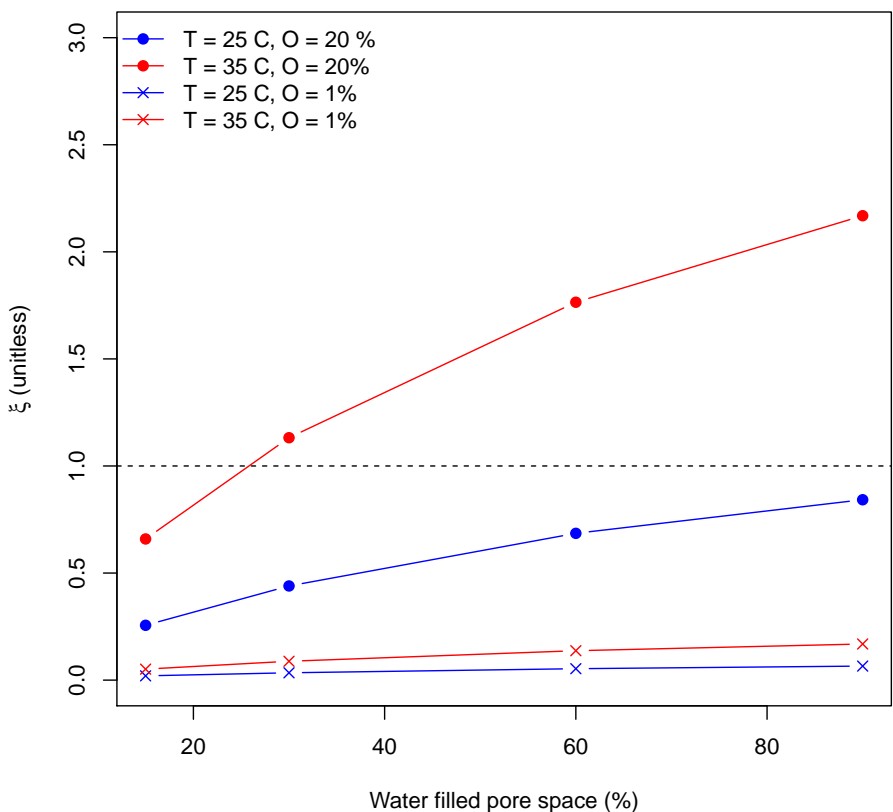

**Figure 5.** Average value of the decomposition modifier $\xi = f(T) \cdot f(M) \cdot f(O)$, predicted for the values of temperature, moisture and oxygen levels applied to all treatment combinations in our experiment. Values above 1 increase decomposition rates and values below 1 reduce them.

though our model has a parsimonious representation motivated partly by the available data, additional details may be included for its use with field observations. For instance, the additional functions in the DAMM model used to represent pore space from bulk density and temperature controls on the $K_X$ terms (Davidson et al., 2012) can help to capture additional complexity under field conditions that are not necessarily relevant under laboratory conditions. The DAMM model, or a variant of it, can be used to represent the term $\xi(t)$. Additionally, extra complexity due to higher heterogeneity of organic matter pools, stabilization and destabilization mechanisms, interactions among microbial enzymes and substrates, and vertical transport, can be incorporated into a larger set of differential equations (e.g. equation 1) dynamically modifying a set of state variables (Sierra and Müller, 2015).

## 5 Conclusions

Based on the experimental data for this boreal soil and the model used, we conclude that decomposition rates can be high i) at high temperatures provided moisture and oxygen levels are not limiting, and ii) at high moisture levels provided oxygen concentrations are not limiting. We found no declines in decomposition rates at high temperatures as predicted by the MMRT; however, the lack of more temperature treatments in a wider temperature range may have been a limitation to find such theoretical optimum. Instead of a decline of respiration rates as promoted by increases in temperature alone, we found important interactions with soil moisture and oxygen concentrations that resulted in declines of respiration when these two variables were limiting. At the level of a soil core with simultaneous changes in moisture levels, strong interactions among temperature, moisture and oxygen levels may override predictions on the temperature dependence at the scale of individual enzymes-substrate pairs (Hobbs et al., 2013; Schipper et al., 2014). These interactions exert a strong control on decomposition, and simultaneous changes of these variables under field conditions should determine the overall rate of decomposition in soils.

## 6 Code and data availability

Code and data necessary to reproduce all results from this manuscript are provided in the supplementary material. Furthermore, the soil incubation dataset used here is part of the soil incubation database (sidb) available as repository in GitHub (https://github.com/SoilBGC-Datashare/sidb).

*Competing interests.* The authors declare that they have no conflict of interest.

*Acknowledgements.* We would like to thank M. SanClements for assistance in sample collection and K. Kluber for preparation of the incubation system. Funding was provided by the Max Planck Society and the German Research Foundation (DFG) through the Emmy Noether Program (SI 1953/2-1). HWL acknowledges the National Science Foundation (NSF) for on-going support. NEON is a project sponsored by the NSF and managed under cooperative support agreement (EF-1029808) to Battelle Inc. Any opinions, findings, and conclusions or recommendations expressed in this material are those of the authors and do not necessarily reflect the views of our sponsoring agency. This paper would not have taken shape if it were not for meaningful engagement with community members.

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
