# Peer review of "Interactions among temperature, moisture, and oxygen concentrations in controlling decomposition rates in a boreal forest soil"

_Biogeosciences, 2016_

## Referee Comment (RC1) · L. Schipper (Referee) · 12 Nov 2016

This paper reports a relatively simple factorial experiment of soil respiration response of moisture, temperature and oxygen. This is an important topic if we are to accurately model respiration of soils in temporally and spatially variable environments. One might think that these relationships have been well constrained already but when trying to find specific examples in the literature it is not easy to find many examples. A simplification of the DAMM model is used to explore data. A nice addition is inclusion of an oxygen treatment to distinguish between the role of water in controlling oxygen supply and carbon diffusion. The paper is easy to read and follow and I generally have few comments. I am not really expert in modelling side of soil carbon dynamics and will

limit my comments here.

Specific comments 1. While a high C content soil was supposedly selected to avoid carbon limitation during the incubation this does not mean that the labile fraction of C would not be depleted. This is important as it is possible for depletion of labile C occurs faster at higher temperatures. The authors can check whether this might have occurred by examining the timeline of $CO_2$ production – if carbon supply was not limiting then respiration rates should be linear and not reach a plateau. Do authors have this information? Currently reporting only the total $CO_2$ after 35 days.

2. Alternatively, a rise in rate through time would indicate adaptation and/or microbial growth during the incubation. Are the authors confident during the 35 days that microbial adaption to constant moisture, temperature and oxygen conditions has not occurred? If this does occur then the model fitting data between different microbial populations. The authors need to acknowledge these possibilities and present some information or rationalisation to overcome them.

3. What was the temperature range in the field that the soils are exposed to?

4. What bulk density was the soil packed to in the cores? Do these represent what might be observed in the field?

5. Pg 3 8 is 'fallowed' meant to be 'followed'?

6. Pg 4 ln 5-10 Include abbreviations O, Ko, W in text

7. Pg 4 ln 23 There were only two temperatures used so that statement respiration did not decrease at higher temperatures should strictly be singular "at the higher temperature".

8. What are the error bars on fig 2? Fig2 I also printed this out in black and white and it was very difficult to see what line was what, symbol could be changed and a dashed line used.

9. Figure 2 and 3 this not really my area and I think a little more description of what these graphs mean would be useful.

10. Figure 4 is it reasonable to make prediction of a full curve of temperature response based two temperatures? And furthermore make prediction above and below the temperature measured? Similarly a very steep curve is predicted for the oxygen content response between two end points.

11. Pg 7 ln 5. I disagree with the statement of increases in temperature being almost always associated with decreases in soil moisture is really a matter of temporal scale of interest. For example between seasons this is certainly possible wet and cold vs hot and dry and this would allow microbes time to adapt. But increases in temperature diurnally can also occur. It would be unusual for a soil to cycle by 5 to 10 C during a 24 hour period where moisture content would be steady and there is less time for adaptation.

12. Pg 9 Conclusions and discussion. That the authors did not find a decline in respiration at a single higher temperature (35C) but this does not mean that MMRT or similar functions are not important in moderating microbial responses in soil. The authors only had two temperatures 25 and 35 C. For the respiration rate to be lower at 35C than 25 C would require the temperature optimum (temperature at which the respiration rate is maximal) to be closer to 25C than 35C. If a temperature optimum for soil respiration was near or greater than 35 C there would be no observed decline in respiration in comparison to the rate at 25C. If I have my logic correct then there is no support for the argument in the conclusions that scale in this case matter with respect to extrapolating MMRT from enzymes to soil systems.

---

## Referee Comment (RC2) · Anonymous Referee #2 · 12 Dec 2016

This manuscript addresses the response of decomposition to three important environmental factors that are closely linked to global change in a very comprehensive way, by performing a multifactorial laboratory study on C rich Arctic soils using a broad range of temperatures, moisture and oxygen concentrations. Even if the effect of these three single factors on decomposition has been extensively studied, combined responses are not yet well understood. A better understanding of interactions between effects of these factors on decomposition is important, as global change is expected to affect all three factors simultaneously in many regions during the next decades. Moreover, changes in decomposition rate can exert strong feedback mechanisms to the climate system, especially at high northern latitudes, which contain both the highest amounts

of soil C and are exposed to the strongest warming.

The conclusions from this research are not highly surprising ((1) temperature stimulates decomposition, but only if moisture and oxygen are available in sufficient concentrations and (2) the oxygen limitation is the main cause for decreasing decomposition rates at high moisture levels). However, a comprehensive study, investigating a broad range of moisture and oxygen levels at different warming levels was still lacking, and gives a more solid base for projections. The manuscript is well written and concise and is easy to follow.

Below, I list my few comments and suggestions on the manuscript:

p. 3, line 5:

The soil columns contained 450 g of homogenized soil. It would also be good to have an idea of the dimensions of the columns (diameter, height). This determines, for instance, the surface area that is subjected to drying and the distance that the oxygen flow travels through the sample. Further, an estimate of the bulk density of the soil or of the proportion of pore space in the samples would be helpful. It is especially important that the pore space was similar for all soil columns so that differences in diffusion potential of oxygen, water and temperature does not influence the results.

p. 3, lines 9-11:

One of the reason to choose Arctic soils that is mentioned is the "low temperatures at which its microbial community is constantly exposed", which "facilitates the possibility of observing strong responses at the extreme of the temperature range". I agree that one would expect a strong response of the microbial activity after step-increasing the temperature by $\sim$20 to 40°C, but the reaction might be more related to stress physiology than an actual temperature response, especially during such a short treatment period (35 days). Therefore, I would not stress this point too much and briefly touch the issue with stress responses after drastic step-change in environmental factors in

the discussion section.

Additionally, I suggest to add another advantage of using Arctic soils for this incubation study: The large amounts of C stored in the Arctic region in combination with the fast warming (compared to the global average). Also moisture (and oxygen) is an issue in that region because of the impenetrable permafrost layer that is present under a large part of the surface.

Further, it is important to restrain your conclusions to Arctic soils, as their dynamics might differ from soils from more moderate or tropical climates. For instance, it has recently been shown that the C balance of soils from Arctic and subarctic regions are more sensitive to warming. It might be that the influence of moisture and or oxygen (and especially the interactions) differ between climates (and probably soil types, but it would dilute the story too much to dig deeper into this). It would be very interesting to perform a similar study with soils from different climate regions.

p. 4, line 4-6 and 24-25:

The fractionation of slow and fast cycling C pools (with different decomposition rates) is not well introduced. Add a paragraph in the introduction as rationale why it is interesting to separate into slow and fast cycling pools when investigating temperature, moisture and oxygen effects on decomposition rates. Also, expand the discussion on this subject.

p. 4, line 4-6:

Define T, W and O.

I would also change W (Water) into M (Moisture), which fits better with the title.

p. 4, line 21 and 22:

35, 90, 20 and 25, 15, 1: Add units to the numbers.

p. 6, line 3-4:

"Decomposition rates were highly sensitive at a narrow part of the oxygen range, while for moisture this range was wider (Figure 4)."

The oxygen range in this study covered the full range of oxygen that can be expected, from 1% (anoxic) to 20% (the maximum that can be expected; atmospheric O2 concentration). The range with the highest sensitivity to oxygen in Figure 4 runs from 0 to 2.5%, which is about 12.5% of the range.

Also for moisture a broad range is covered (15 to 90%). The range with the highest sensitivity to oxygen in Figure 4 runs from 0 to 10%, which is about 11% of the range (if the maximum is set to 90%).

As there is little difference between 12.5 and 11% of the range, I do not understand the statement that the sensitivity to moisture occurred at a broader range. Can you explain this in more detail?

p. 8, Figure 4:

It is strange that the strongest response for all three parameters occurs outside the treatment range of this study. Is it possible to extrapolate your findings that far?

p. 1 line 7; p. 3 Lines 9; p. 9 line 8

Change "arctic" into "Arctic"

———————————————

---

## Referee Comment (RC3) · L. Schipper (Referee) · 12 Dec 2016

Apologies, in last sentence of comment 11, I left out the word "not" so the comment should read:

It would be not unusual for a soil to cycle by 5 to 10 C during a 24 hour period where moisture content would be steady and there is less time for adaptation.

---

## Referee Comment (RC4) · Anonymous Referee #3 · 15 Dec 2016

General comments.

Interactive controls of multiple environmental factors on the decomposition of soil organic matter, and its loss to the atmosphere, remains a challenging research question despite much effort by the research community. This study presents a useful factorial experimental manipulation of temperature, water, and most interestingly [O2], to evaluate the interactive controls of these factors on C loss in from boreal forest soils. The choice of soils probably limited the effect of substrate limitation, and also represents a pool of C (boreal forest soil C) that is potentially a major contributor to future feedbacks between climate and the terrestrial C cycle. The data-model integration is a useful effort, and I like that the authors present the data and code, as it will very likely be

useful to other researchers. However, I think the paper needs to present the modeling approach and the results somewhat more clearly. Also, the results and discussion seem a little sparse on several of the key questions the authors introduce. The writing is clear, although I think the paper contains several ambiguities related to the brevity. Some features need to be described in more detail.

Specific comments.

P1 Line 7: The site is boreal, not arctic.

P1 Line 9: The conclusion about temperature effect need to be tempered or qualified in the context of the limited range of temperatures evaluated.

P1 Line 10: This is a significant conclusion, even though it seems relatively obvious-having a good experimental design to say this conclusively is useful.

P2 Line 4: How does this 45C threshold correspond to your high temperature? Is 45C broadly constant across ecosystems?

P2 Line 24: True, and a major strength of this study.

P3 Line 8: Again, boreal, not arctic.

P3 Line 9: This statement is not necessarily true depending on the content of labile, readily respired substrate. It should be explored a little further and contextualized with other studies that evaluate substrate limitation of soil respiration in organic soils, especially from boreal regions.

P3 Line 18: It is unclear exactly how this measurement was used to evaluate the soil respiration or decomposition rate. Can you please clarify? Was it evaluated as change over a set time interval, or as increase over the known background from the input air? At what frequency was this measured?

Eqn 1: Please be clear about what exactly dC/dt represents. Is it the instantaneous or the cumulative $dCO_2$, is it $CO_2$ or $CO_2$-C?

[Figure]

P4 Line 10: Does this mean that gamma also varies by each treatment level? And initial C1 and C2 also vary by treatment level? I would like to see some presentation of the actual C fluxes, and the change in C1 and C2 over time.

P4 Line 12: Fitting the full model in eqn 2, is gamma now fixed? Also, are there limitations to fitting a q10 function with only two temperature points?

P4 Line 14: Thanks for presenting this supplement.

P 4 Line 27: I am more surprised at how similar the k1 and k2 values in Fig 2 are across such a broad range of O2: Can you explain this result more clearly.

P5 Line 5: Can you please elaborate a little further on fig 3? We do see a few seemingly high correlations that might be worth describing in more detail. For instance, Ko and ks.

P5 Line 6: Am I missing the posterior parameter estimates? It would be very useful to have a table of these parameter values and credible intervals.

P5 Line 7: Why did you use this range rather than the 2.5-97.5? It looks like your estimates of the temperature function might be challenging in that case, which isn't that surprising with only two temps. I think it is worth revising these figures to have both the 25-75 and then standard 95% credible interval presented.

P7 Line 1: The discussion should give some analysis of the temperature response. In particular, how do the estimated q10 values compare to other q10 values using soils from similar boreal forest sites? Also, what is the temperature range at this site? You describe the 45C threshold in the introduction, but then use a much lower temperature as the high temp. Is this higher than temperatures the soil organisms at this site regularly experience? Is it higher than projected future temperatures for this site?

P8 Line 3: Can you please describe some of these interactions more specifically? It seems as though a lot of the work is presenting marginal responses. Is there some reduction of temperature sensitivity at high water content? or a reduction of oxygen

sensitivity at low water content? Please clarify what interactions you mean.

P8 Line7: I am not sure I follow. Looking at figure 1, most CO2 was respired at high water content. Are you thus comparing the low to the high oxygen rates, and then inferring a response in the absence of continuous oxygen flow? Please be explicit about that.

P9 Line 3: Perhaps discuss more thoroughly how the inclusion of dynamically changing air-filled pore space might relate to your results. This is a suggestion, not a necessary revision.

Technical corrections

P1 Line 16: *significantly

---

## Referee Comment (RC5) · Anonymous Referee #4 · 3 Jan 2017

Sierra et al studied how the interactions among temperature, moisture and oxygen concentrations control the decomposition rates of soil organic matter using a combination of modeling and incubations. The study is of course important, however, the paper could be further improved if clarification is done for a few places that I will list point by point below.

First, there is some confusion in describing the level off of decomposition rates at high temperatures. E.g. at P2 L6, enzyme denature should be described as irreversible enzyme denature, so one will not confuse it with reversible enzyme denature. As a mater of fact, the MMRT theory is largely based on reversible enzyme denature (though its authors did not say so), which was known as early as in the 1980s (Murphy et al., 1990:

Common features of protein unfolding and dissolution of hydrophobic compounds, Sciences). The idea was then combined with the concept of a single rate-limiting "master reaction" to model the respiration of bacteria by Ratkowsky et al. (2005: J of Theoretical Biology). A much earlier study by Sharpe and DeMichele (1977: J. Theoretical Biology) also derived a similar curve as MMRT, and was used in the model ECOSYS (Grant et al, 1993: Soil Biol. Biochem.) to simulate microbial decomposition. More recently, the same idea was applied in the model Tang and Riley (2015: Nature Climate Change). I think the authors of this study should report these developments so readers will have a more complete picture of this problem.

Second, P2. L10-11, I think this criticism is not quite true. Authors who applied these concepts never said moisture should remain constant; rather they just focused on temperature, because temperature is considered as the most important factor. Moisture effect could be very well incorporated into those applications, which may be under way and ECOSYS has done this in the 1990s.

Third, in describing the moisture effect, the authors missed the physiological effect that the moisture will impose on microbes as soil matric pressure becomes more negative. Such effect was shown to be important in Grant and Rochette (1994: Soil Sci. Soc. Am. J.), Manzoni et al. (2016: Soil Biology and Biochemistry) and Yan et al. (2016: Biogeochemistry).

Fourth, in describing the incubation, the geometry of the incubated soil is not clear, e.g. what is the thickness of the cylindrical soil column? Such overall thickness will definitely affect the interpretation of the empirical data.

Finally, in describing the modeling approach, the authors did not lay out the hypotheses that lead to their model structure. For instance, under what conditions should this model structure be assumed applicable? Apparently, the model as proposed will only be useful for a soil column neither too shallow nor too deep. For a too shallow soil in natural environment, oxygenation will be very effective under the variable environment

(through mechanisms such as wind pumping), so both the oxygen and moisture effect will be hard to discern from empirical data. For a too deep soil, difference in the vertical distribution of all decomposition variables will invalidate the homogenous assumption as built in the model. Also, the model assumes the microbial dynamics is totally slaved to the moisture and oxygen effects, so hysteretic behavior due to population dynamics as identified in Tang and Riley (2015) will be missing. The population dynamics may be very important in field conditions.

Other comments:

P3 L29-32: this could be summarized as parametric equifinality.

---

## Referee Comment (RC6) · L. Schipper (Referee) · 12 Jan 2017

Comment on reviewer 4

As the author/co-author of the original papers on MMRT I would like to address a comment made by reviewer 4 on what MMRT is representing.

The reviewer makes the comment: "As a matter of fact, the MMRT theory is largely based on reversible enzyme denature (though its authors did not say so), which was known as early as in the 1980s (Murphy et al., 1990: C1 BGD Interactive comment Printer-friendly version Discussion paper Common features of protein unfolding and dissolution of hydrophobic compounds, Sciences)."

[Figure]

This is not correct. We are aware of earlier work on reversible denaturation. In our early papers on MMRT, we concurrently measured unfolding rates (denaturation) and reaction kinetics and clearly show that unfolding rates (whether reversible or not) were very minor contributors to the initial reductions in enzyme rate as temperature increased (Hobbs et al. 2013). In many cases, enzyme rates decline at temperatures well below the temperature for unfolding and MMRT accounts for this in the absence of unfolding (reversible or irreversible). Denaturation, of course, does occur at higher temperature but is not responsible for the initial decline in temperature sensitivity. We have a specific section in one of our more recent papers that addresses denaturation and we make that point that MMRT is independent of denaturation/unfolding (Arcus et al. 2016). MMRT actually describes how the activation energy changes with temperature, driven by the change in heat capacity between the enzyme-substrate complex and the enzyme-transition state complex. MMRT is simply a theoretical extension of transition state theory (Eyring and Polyani) as it is applied to enzyme kinetics (M. Garcia-Viloca, J. Gao, M. Karplus, D. G. Truhlar, How enzymes work: analysis by modern rate theory and computer simulations. Science. 303, 186–195 (2004))..

We also note that if denaturation was responsible for defining the temperature optimum then for many soil biological processes (e.g., methanogenisis, methane oxidation, soil enzymes) would have very high denaturation rates at temperatures less than $30-40$ degrees (Schipper et al. 2014). In our recent paper describing MMRT, we make evolutionary arguments for low values of Topt without invoking low denaturation temperatures for individual enzymes.

Other curved functions have certainly been developed but these curves are empirical equations and generally considers denaturation (reversible or not) as the mechanism, while MMRT is theoretically derived, is independent of denaturation and has recently been validated by direct measurements of enzyme heat capacity changes on the reaction coordinate (Firestone et al 2016).

References Arcus VL, Prentice E, Hobbs JK, Mulholland AJ, Vander Kamp MW, Pudney CR, Parker EJ, Schipper LA (2016) On the temperature dependence of enzyme-catalysed rates. Biochemistry 55: 1681-1688

Hobbs JK, Jiao W, Easter AD, Parker EJ, Schipper LA, Arcus VL (2013) Change in heat capacity for enzyme catalysis determines temperature dependence of enzyme catalyzed rates. ACS Chemical Biology 8(11): 2388-2393

R. S. Firestone, S. A. Cameron, J. M. Karp, V. L. Arcus, V. L. Schramm, Heat Capacity Changes for Transition-State Analogue Binding and Catalysis with Human 5'-Methylthioadenosine Phosphorylase. ACS Chem. Biol. (2016), doi:10.1021/acschembio.6b00885.

Schipper LA, Hobbs JK, Rutledge S, Arcus VL (2014) Thermodynamic theory explains the temperature optima of soil microbial processes and high Q(10) values at low temperatures. Global Change Biology 20(11): 3578-3586

Sincerely Louis Schipper and Vickery Arcus

---

## Author Comment (AC1) · 23 Jan 2017

We thank reviewer 1 for his comments and for actively engaging in the discussion motivated by this manuscript (see comments to Reviewer 4). Here we quote his comments in *italics* and provide our answers below each major comment.

*This paper reports a relatively simple factorial experiment of soil respiration response of moisture, temperature and oxygen. This is an important topic if we are to accurately model respiration of soils in temporally and spatially variable environments. One might think that these relationships have been well constrained already but when trying to find specific examples in the literature it is not easy to find many examples. A simplification of the DAMM model is used to explore data. A nice addition is inclusion of an*

[Figure]

*oxygen treatment to distinguish between the role of water in controlling oxygen supply and carbon diffusion. The paper is easy to read and follow and I generally have few comments. I am not really expert in modelling side of soil carbon dynamics and will limit my comments here.*

We think one of the reasons there's relatively little work on this subject is because the difficulties in controlling three factors simultaneously. Full factorial experiments involving more than two factors are not so common in ecology even though they can help us to better understand multiple factor interactions among environmental drivers.

*Specific comments 1. While a high C content soil was supposedly selected to avoid carbon limitation during the incubation this does not mean that the labile fraction of C would not be depleted. This is important as it is possible for depletion of labile C occurs faster at higher temperatures. The authors can check whether this might have occurred by examining the timeline of CO2 production – if carbon supply was not limiting then respiration rates should be linear and not reach a plateau. Do authors have this information? Currently reporting only the total CO2 after 35 days.*

We do have this information. Cumulative $CO_2$ production from all treatments is presented in the Figure 1 below.

The treatments with the highest amounts of respired $CO_2$ showed almost linear increases in respiration and provide no evidence of a depletion of labile carbon. Treatments with low water filled pore space (WFPS) and low oxygen levels show a tendency to reach a plateau, but this is likely due to a strong decrease of respiration rates due to water and oxygen limitation and not due to substrate depletion. The soil is the same for all treatments so we expect labile carbon to be the same in all cases.

The results from the model optimization shows that the fraction of fast 'labile' carbon is around 56% of the total initial carbon in the incubations (Table 1 in new version). This is common for these highly organic soils and it is highly unlikely that this labile carbon is depleted during a 35-day incubation.

*2. Alternatively, a rise in rate through time would indicate adaptation and/or micro-bial growth during the incubation. Are the authors confident during the 35 days that microbial adaption to constant moisture, temperature and oxygen conditions has not occurred? If this does occur then the model fitting data between different microbial populations. The authors need to acknowledge these possibilities and present some information or rationalisation to overcome them.*

The cumulative respiration data in Figure 1 below, presents clear and distinct trends for each of the treatments. Treatments with high levels of temperature, moisture and oxygen show near linear trends, which is an indication that the microbial communities are growing at an almost constant rate and are not experiencing any resource limitation. In treatments with low resource levels (moisture and oxygen), microbial growth declines during incubation time suggesting depletion of resources necessary for growth. We do not believe that the linear trend in high-resource treatments is an indication of micro-bial adaptation that somehow would invalidate our results. On the contrary, this is a strong indication that growth, and therefore respiration rates, is not limited by the levels of available resources. More importantly, the slope of the near-linear increases in these treatments is highly dependent on the factor levels, a strong indication that rates depend on the three main factors imposed, something that is later backed up by our modeling analysis.

*3. What was the temperature range in the field that the soils are exposed to?* This soil was collected from the active layer at a Caribou/Poker Creek watershed in central Alaska. The active surface layer is exposed to large changes in temperature during the year, from a minimum temperature close to -19°C to a maximum of 18°C, so the annual temperature range is close to 40°C. This information was obtained from the Bonanza Creek LTER site, data summaries for the CPCRW station (http://bnznet.iab.uaf.edu/vdv/vdv_historical.php). We included this information in the site description section of our manuscript.

*4. What bulk density was the soil packed to in the cores? Do these represent what*

[Figure]

*might be observed in the field?*
Each cylinder had a volume of 1570 cm$^3$ (10 cm diameter, 20 cm height), and contained 450 g of soil. About half of the cylinder was filled with soil, so the approximate bulk density was 0.57 g cm$^{-3}$. Typical values for bulk density in organic horizons and peats are between 0.1 and 0.5 g cm$^{-3}$ (Hossain et al., 2015). We believe the bulk density within our cylinders was realistic and corresponds to typical values for these type of soils.

*5. Pg 3 8 is 'fallowed' meant to be 'followed'?*
Changed.

*6. Pg 4 ln 5-10 Include abbreviations O, Ko, W in text*
Done.

*7. Pg 4 ln 23 There were only two temperatures used so that statement respiration did not decrease at higher temperatures should strictly be singular "at the higher temperature".*
Changed as suggested.

*8. What are the error bars on fig 2? Fig2 I also printed this out in black and white and it was very difficult to see what line was what, symbol could be changed and a dashed line used.*
Arrows represent 25 and 75% quantiles of the distribution of the parameters obtained through Bayesian optimization. This information was added to the figure caption. Symbols in figure were changed to improve readability in white-and-black print outs.

*9. Figure 2 and 3 this not really my area and I think a little more description of what these graphs mean would be useful.*
We added more description on the main text and on the figure captions.

*10. Figure 4 is it reasonable to make prediction of a full curve of temperature response based two temperatures? And furthermore make prediction above and below the tem-*

*perature measured? Similarly a very steep curve is predicted for the oxygen content
response between two end points.*

Good point. Here we only wanted to show the predictions from a model that fitted well
the data at the specific points within possible ranges for the controlled variables. We
believe it is also interesting to know what the model predicts within and outside the
range of possible values. However, one must be very careful with the interpretation of
these results since, as the reviewer points out, we do not have data outside the specific
points where we imposed our treatments.

To address this issue we modified Figure 4, showing explicitly predictions for the spe-
cific treatment combinations where we have data, and only plotting the predicted curves
as dashed lines for reference.

*11. Pg 7 ln 5. I disagree with the statement of increases in temperature being almost
always associated with decreases in soil moisture is really a matter of temporal scale
of interest. For example between seasons this is certainly possible wet and cold vs
hot and dry and this would allow microbes time to adapt. But increases in temperature
diurnally can also occur. It would be not unusual for a soil to cycle by 5 to 10 C during
a 24 hour period where moisture content would be steady and there is less time for
adaptation.*

Yes, this is a matter of scale. At some time scales, increases in temperature are accom-
panied with increases in moisture (e.g. as soil unfreezes), it may dry at other scales
(e.g. in the spring season in mediterranean ecosystems), or it may remain constant
during a 24 hour cycle. We explored these different dynamics in a previous manuscript
(Sierra et al., 2015, Fig 1), and concluded that in a large number of relevant cases, soil
temperature and soil moisture change simultaneously.

*12. Pg 9 Conclusions and discussion. That the authors did not find a decline in
respiration at a single higher temperature (35C) but this does not mean that MMRT
or similar functions are not important in moderating microbial responses in soil. The
authors only had two temperatures 25 and 35 C. For the respiration rate to be lower*

[Figure]

*at 35C than 25 C would require the temperature optimum (temperature at which the respiration rate is maximal) to be closer to 25C than 35C. If a temperature optimum for soil respiration was near or greater than 35 C there would be no observed decline in respiration in comparison to the rate at 25C. If I have my logic correct then there is no support for the argument in the conclusions that scale in this case matter with respect to extrapolating MMRT from enzymes to soil systems.*

The reviewer is right here, and we acknowledge that our logic was flawed. It is correct that our two temperature treatments may not be enough to observe any potential decline in respiration rates as predicted by the MMRT, and this has little to do with scales. However, we still believe that there's an important mismatch between the scale at which the MMRT operates and the scale for observing soil respiration in soil cores and soil pits. The MMRT predicts a decline in respiration at high temperature provided all other environmental factors remain constant. But in most cases in soils, both temperature, moisture and oxygen change simultaneously. Our point with the manuscript is to bring to the attention that these interactions are very important for predictions at the soil core level, even though there may be mismatches with the predictions at the enzyme level.

To address this comment we re-wrote the conclusion section and are now more precise about the limitations of our experiment and the mismatch among scales.

**References**

Hossain, M., Chen, W., and Zhang, Y. (2015). Bulk density of mineral and organic soils in the Canada's arctic and sub-arctic. *Information Processing in Agriculture*, 2(3–4):183 – 190.

Sierra, C. A., Trumbore, S. E., Davidson, E. A., Vicca, S., and Janssens, I. (2015) Sensitivity of decomposition rates of soil organic matter with respect to simultaneous changes in temperature and moisture, Journal of Advances in Modeling Earth Systems, 7, 335–356.

[Figure]

[Figure]

**Fig. 1.** Cumulative respiration for all treatments in the incubation experiment.

[Figure]

---

## Author Comment (AC2) · 23 Jan 2017

We thank reviewer 2 for his/her comments on our manuscript. Here we quote comments in *italics* and provide our answers below each major comment.

*p. 3, line 5: The soil columns contained 450 g of homogenized soil. It would also be good to have an idea of the dimensions of the columns (diameter, height). This determines, for instance, the surface area that is subjected to drying and the distance that the oxygen flow travels through the sample. Further, an estimate of the bulk density of the soil or of the proportion of pore space in the samples would be helpful. It is especially important that the pore space was similar for all soil columns so that differences in diffusion potential of oxygen, water and temperature does not influence the results.*

[Figure]

For each cylinder, we had 450 g of soil in a volume of about 785 cm$^3$, which results in a bulk density of about 0.57 g cm$^{-3}$. This was similar for all samples and care was taken to have similar bulk density across all treatments.

It is important to keep in mind that soil water modifies the amount of filled pore space, and for this reason our moisture treatment is expressed in water-filled pore space. This obviously has consequences on the diffusion characteristics for each moisture treatment, which results in the observed differences in respiration rates.

*p. 3, lines 9-11: One of the reason to choose Arctic soils that is mentioned is the "low temperatures at which its microbial community is constantly exposed", which "facilitates the possibility of observing strong responses at the extreme of the temperature range". I agree that one would expect a strong response of the microbial activity after step-increasing the temperature by ~20 to °40C, but the reaction might be more related to stress physiology than an actual temperature response, especially during such a short treatment period (35 days). Therefore, I would not stress this point too much and briefly touch the issue with stress responses after drastic step-change in environmental factors in the discussion section. Additionally, I suggest to add another advantage of using Arctic soils for this incubation study: The large amounts of C stored in the Arctic region in combination with the fast warming (compared to the global average). Also moisture (and oxygen) is an issue in that region because of the impenetrable permafrost layer that is present under a large part of the surface. Further, it is important to restrain your conclusions to Arctic soils, as their dynamics might differ from soils from more moderate or tropical climates. For instance, it has recently been shown that the C balance of soils from Arctic and subarctic regions are more sensitive to warming. It might be that the influence of moisture and or oxygen (and especially the interactions) differ between climates (and probably soil types, but it would dilute the story too much to dig deeper into this). It would be very interesting to perform a similar study with soils from different climate regions.*

These are all good points. The reviewer is correct in that the high temperature treatments we applied may induce physiological stress in microbes. The effect may be expressed as a short-term physiological response or as a long-term change in microbial communities (Schimel et al., 2007). Our $CO_2$ respiration measurements however, cannot distinguish between these two type of responses. Although our incubations were short (35 days), this is still enough time for the microbial community to shift. We may not be able to say anything here about the mechanistic response at the microbial level, but we observed an aggregate response that may combine both microbial physiology and community composition. At the level of abstraction we are focusing in this manuscript, this overall response is important for representing climate change effects in soil models.

We also agree with the reviewer in that it would be very interesting to replicate this experiment for soils from different ecosystems. We may be able to observe very different responses for tropical or temperate systems.

We added some text to the methods and discussion section addressing these points.

*p. 4, line 4-6 and 24-25: The fractionation of slow and fast cycling C pools (with different decomposition rates) is not well introduced. Add a paragraph in the introduction as rationale why it is interesting to separate into slow and fast cycling pools when investigating temperature, moisture and oxygen effects on decomposition rates. Also, expand the discussion on this subject.*
This is an important point that we did not address properly in our previous version. It is not only the interaction among multiple environmental factors, but also how they affect different rates. Our modeling approach includes both a fast and a slow pool that are modified by these different environmental factors. We included some text in the introduction, the model description, and the discussion addressing this topic.

*p. 4, line 4-6: Define T, W and O. I would also change W (Water) into M (Moisture), which fits better with the title.*
Definitions were added, and $W$ was changed for $M$ as suggested.

*p. 4, line 21 and 22: 35, 90, 20 and 25, 15, 1: Add units to the numbers.*
Done

*p. 6, line 3-4: "Decomposition rates were highly sensitive at a narrow part of the oxygen range, while for moisture this range was wider (Figure 4)." The oxygen range in this study covered the full range of oxygen that can be expected, from 1% (anoxic) to 20% (the maximum that can be expected; atmospheric O2 con- centration). The range with the highest sensitivity to oxygen in Figure 4 runs from 0 to 2.5%, which is about 12.5% of the range. Also for moisture a broad range is covered (15 to 90%). The range with the highest sensitivity to oxygen in Figure 4 runs from 0 to 10%, which is about 11% of the range (if the maximum is set to 90%). As there is little difference between 12.5 and 11% of the range, I do not understand the statement that the sensitivity to moisture occurred at a broader range. Can you explain this in more detail?*
Our previous description of the intrinsic sensitivities was ambiguous as noted by the reviewer. We re-wrote completely this paragraph in light of other reviewer's comments and the modification of Figure 4. We do not refer here about these ranges anymore, but mostly to the overall shapes of the dependence and sensitivity curves. We also put more emphasis on the predictions for the specific treatment levels and not so much on specific predictions outside the values for which we have no data.

*p. 8, Figure 4: It is strange that the strongest response for all three parameters occurs outside the treatment range of this study. Is it possible to extrapolate your findings that far?*
Figure 4 was modified to avoid emphasis on interpretations outside the ranges where we do not have data. The new version of Fig 4 gives more emphasis on the specific treatment levels where we have measurements, and we only provide model predictions outside these values as a reference for the functional response of the model.

*p. 1 line 7; p. 3 Lines 9; p. 9 line 8 Change "arctic" into "Arctic"*
As Reviewer 3 pointed out, the site is really a boreal and not an Arctic ecosystem. We changed arctic to boreal throughout the manuscript.

**References**

Schimel, J., Balser, T. C., and Wallenstein, M. (2007). Microbial stress-response physiology and its implications for ecosystem function. *Ecology*, 88(6):1386–1394.

---

## Author Comment (AC3) · 23 Jan 2017

We thank reviewer 3 for his/her comments on our manuscript. Here we quote comments in *italics* and provide our answers below each major comment.

*P1 Line 7: The site is boreal, not arctic.*
Yes, we changed to boreal.

*P1 Line 9: The conclusion about temperature effect need to be tempered or qualified in the context of the limited range of temperatures evaluated.*
We modified this sentence slightly, not mentioning that decomposition 'increases' with 'increases' in temperature, since we only have two temperatures; but mentioning that

decomposition rates 'were high' at high temperatures provided oxygen and moisture were not limiting.

*P1 Line 10: This is a significant conclusion, even though it seems relatively obvious-having a good experimental design to say this conclusively is useful.*
Thanks.

*P2 Line 4: How does this 45C threshold correspond to your high temperature? Is 45C broadly constant across ecosystems?*
We mention this 45°C threshold only to introduce the MMRT, which is supposed to operate at lower temperatures than this threshold for enzyme denaturation.

*P2 Line 24: True, and a major strength of this study.*
Thanks.

*P3 Line 8: Again, boreal, not arctic.*
Changed.

*P3 Line 9: This statement is not necessarily true depending on the content of labile, readily respired substrate. It should be explored a little further and contextualized with other studies that evaluate substrate limitation of soil respiration in organic soils, especially from boreal regions.*
We believe that high organic soils minimize the potential of substrate limitation, but the reviewer is right in that this may not be the case always. For clarity, we added the word 'may' to this sentence.

*P3 Line 18: It is unclear exactly how this measurement was used to evaluate the soil respiration or decomposition rate. Can you please clarify? Was it evaluated as change over a set time interval, or as increase over the known background from the input air? At what frequency was this measured?*
For each cylinder, fluxes were measured every other day as the difference in concentration between the output and input air, multiplied by the air mass flow rate. We added a

more detailed description about the quantification of respiration rate to the new version of the manuscript.

*Eqn 1: Please be clear about what exactly dC/dt represents. Is it the instantaneous or the cumulative dCO2, is it CO2 or CO2-C?*
Here, $dC/dt$ represents the instantaneous change in the carbon content for the incubated soil. The respired $CO_2$-C is obtained after solving the system of differential equations and calculating the output flux from the numerical output. We added more details on the model description to make this clear.

*P4 Line 10: Does this mean that gamma also varies by each treatment level? And initial C1 and C2 also vary by treatment level? I would like to see some presentation of the actual C fluxes, and the change in C1 and C2 over time.*
Yes, the values of $\gamma$ change for each treatment level for the first optimization and this is presented in Figure 2. For the second optimization, we obtain a probability distribution for $\gamma$, which is presented in Figure 3.

*P4 Line 12: Fitting the full model in eqn 2, is gamma now fixed? Also, are there limitations to fitting a q10 function with only two temperature points?*
Again, for the second optimization, where equation 2 is set explicitly, we obtain a probability distribution for $\gamma$. It is not a value that changes from one treatment to another, but a range of values with some probability.

The main limitation of fitting a $Q_{10}$ function with two temperature values is that the obtained uncertainty range is very high, which is evident in the probability distribution presented in Figure 3, and the predictions in Figure 4.

*P4 Line 14: Thanks for presenting this supplement.*
Thanks for the comment.

*P 4 Line 27: I am more surprised at how similar the k1 and k2 values in Fig 2 are across such a broad range of O2: Can you explain this result more clearly.*

This is well explained by the sensitivity functions in Figure 4. The intrinsic sensitivity of decomposition rates with respect to temperature is higher for temperature, intermediate for moisture, and lower for oxygen for the treatment levels we selected.

*P5 Line 5: Can you please elaborate a little further on fig 3? We do see a few seemingly high correlations that might be worth describing in more detail. For instance, Ko and ks.*
Here the concept of 'high' correlations is relevant. For exploring collinearity between parameter sets we are interested in finding correlations above 90-95%. This would be indicative that parameter values lie within a straight line. In analyses of ecological data, researchers often describe correlations above 0.3-0.4 as 'high' due to the inherent variability of ecological processes. However, the aim here with Fig 3 is to find near linear correlations as evidence of collinearity among parameter values. Therefore, we do not consider the obtained correlations as high for the purpose of our analysis.

*P5 Line 6: Am I missing the posterior parameter estimates? It would be very useful to have a table of these parameter values and credible intervals.*
We added a table with these parameter values and their uncertainty.

*P5 Line 7: Why did you use this range rather than the 2.5-97.5? It looks like your estimates of the temperature function might be challenging in that case, which isn't that surprising with only two temps. I think it is worth revising these figures to have both the 25-75 and then standard 95% credible interval presented.*
We included now the 5 and 95% interquartile ranges for the second optimization.

*P7 Line 1: The discussion should give some analysis of the temperature response. In particular, how do the estimated q10 values compare to other q10 values using soils from similar boreal forest sites? Also, what is the temperature range at this site? You describe the 45C threshold in the introduction, but then use a much lower temperature as the high temp. Is this higher than temperatures the soil organisms at this site regularly experience? Is it higher than projected future temperatures for this site?*

Our new version of the manuscript includes a discussion on the obtained dependence and sensitivity functions for temperature, moisture, and oxygen, but we do not include a discussion on comparing the obtained $Q_{10}$ values with others found in the literature.

The objective of our modeling analysis was to obtain relevant parameters for the interpretation of the experimental results. We are not concerned here on describing or interpreting our parameter values as representative for modeling this type of soils under non-experimental conditions. Under field-conditions other physical and biological processes may have also a strong effect on decomposition and respiration rates not relevant under the experimental conditions of our experiment. For this reason, we are reluctant to compare the dependence function and the $Q_{10}$ values against others found for other type of soils under completely different measurement, experimental, and modeling setups. Previously, I have strongly criticized the practice of comparing $Q_{10}$ values from different studies using different functions (Sierra, 2012), and do not consider appropriate to do such a comparison here.

*P8 Line 3: Can you please describe some of these interactions more specifically? It seems as though a lot of the work is presenting marginal responses. Is there some reduction of temperature sensitivity at high water content? or a reduction of oxygen sensitivity at low water content? Please clarify what interactions you mean.*
To address these interactions more explicitly we introduce a new figure (Fig 5) calculating the value of $\xi$ for the specific treatment combinations. This figure help us to discuss the interactions among the three variables in more detail.

*P8 Line7: I am not sure I follow. Looking at figure 1, most CO2 was respired at high water content. Are you thus comparing the low to the high oxygen rates, and then inferring a response in the absence of continuous oxygen flow? Please be explicit about that.*
This sentence refers to the study of Tucker and Reed (2016) and not to the results presented in our manuscript.

*P9 Line 3: Perhaps discuss more thoroughly how the inclusion of dynamically changing air-filled pore space might relate to your results. This is a suggestion, not a necessary revision.*
We extended this paragraph to give better details about how the model can be modified for representing more complex processes or for applications to the field level.

*Technical corrections*
*P1 Line 16: \*significantly*
Done

**References**

Tucker, C. L. and Reed, S. C. (2016). Low soil moisture during hot periods drives apparent negative temperature sensitivity of soil respiration in a dryland ecosystem: a multi-model comparison. *Biogeochemistry*, 128(1):155–169.
Sierra, C. (2012). Temperature sensitivity of organic matter decomposition in the arrhenius equation: some theoretical considerations. *Biogeochemistry*, 108(1):1–15.

---

## Author Comment (AC4) · 23 Jan 2017

We thank reviewer 4 for his/her comments on our manuscript. Here we quote comments in *italics* and provide our answers below each major comment.

*First, there is some confusion in describing the level off of decomposition rates at high temperatures. E.g. at P2 L6, enzyme denature should be described as irreversible enzyme denature, so one will not confuse it with reversible enzyme denature. As a mater of fact, the MMRT theory is largely based on reversible enzyme denature (though its authors did not say so), which was known as early as in the 1980s (Murphy et al., 1990: Common features of protein unfolding and dissolution of hydrophobic compounds, Sciences). The idea was then combined with the concept of a single rate-limiting "master*

*reaction" to model the respiration of bacteria by Ratkowsky et al. (2005: J of Theoretical Biology). A much earlier study by Sharpe and DeMichele (1977: J. Theoretical Biology) also derived a similar curve as MMRT, and was used in the model ECOSYS (Grant et al, 1993: Soil Biol. Biochem.) to simulate microbial decomposition. More recently, the same idea was applied in the model Tang and Riley (2015: Nature Climate Change). I think the authors of this study should report these developments so readers will have a more complete picture of this problem.*

It is incorrect to say that the MMRT is based on reversible enzyme denaturation. The answer to this comment by Reviewer 1 clearly explains why this is not the case, and our explanation that MMRT describes the changes in activation energy with temperature is in fact correct. Furthermore, we believe that a discussion on the origins of one enzyme-level theory over another is well beyond the scope of this manuscript. Our measurements and level of abstraction are at the level of overall respiration fluxes and how are they influenced by temperature, moisture and oxygen. We only mention the MMRT and denaturation as a context for expected responses, but a detailed description of enzyme reaction theories would introduce a level of detail that would serve more as a distraction rather than a useful context for the present analysis.

*Second, P2. L10-11, I think this criticism is not quite true. Authors who applied these concepts never said moisture should remain constant; rather they just focused on temperature, because temperature is considered as the most important factor. Moisture effect could be very well incorporated into those applications, which may be under way and ECOSYS has done this in the 1990s.*

This is really not a criticism, but rather an important consideration when using these functions. Temperature effects on enzyme activity and decomposition rates operate under the assumption that all else remains equal except temperature. This is very important for developing and testing these functions, but in practical applications one must also consider that other environmental factors also change. This is the only point we wanted to make here.

*Third, in describing the moisture effect, the authors missed the physiological effect that the moisture will impose on microbes as soil matric pressure becomes more negative. Such effect was shown to be important in Grant and Rochette (1994: Soil Sci. Soc. Am. J.), Manzoni et al. (2016: Soil Biology and Biochemistry) and Yan et al. (2016: Biogeochemistry).*
We added a sentence addressing this physiological effect.

*Fourth, in describing the incubation, the geometry of the incubated soil is not clear, e.g. what is the thickness of the cylindrical soil column? Such overall thickness will definitely affect the interpretation of the empirical data.*
We included a description of the area, height, and volume of the soil columns as well as a calculation of the bulk density of the soils.

*Finally, in describing the modeling approach, the authors did not lay out the hypotheses that lead to their model structure. For instance, under what conditions should this model structure be assumed applicable? Apparently, the model as proposed will only be useful for a soil column neither too shallow nor too deep. For a too shallow soil in natural environment, oxygenation will be very effective under the variable environment (through mechanisms such as wind pumping), so both the oxygen and moisture effect will be hard to discern from empirical data. For a too deep soil, difference in the vertical distribution of all decomposition variables will invalidate the homogenous assumption as built in the model. Also, the model assumes the microbial dynamics is totally slaved to the moisture and oxygen effects, so hysteretic behavior due to population dynamics as identified in Tang and Riley (2015) will be missing. The population dynamics may be very important in field conditions.*
The scope of application of the parameterized model does not go beyond than that of the incubated soils. It is not our objective to propose a general model that can be applied to field conditions. We were only interested in testing a model that include the three main environmental factors (Temperature, Moisture, Oxygen) on a homogeneous organic soil consisting of two kinetic pools. We acknowledge that for predicting field

data a more complex model may be needed, which is expressed in the last paragraph of our discussion.

To make this point even more clear, we added a sentence in our model description section indicating that the objective of this model structure is only to explain our experimental data, but more complex models may be needed for other applications.

*P3 L29-32: this could be summarized as parametric equifinality.*
Yes, these are similar terms. We added the word 'equifinallity' in parenthesis so readers know that they are synonymous.